# Data and Model Harmonization Research Challenges in a Nation Wide Digital Twin

Jean-Sébastien Sottet *,† and Cédric Pruski †

Luxembourg Institute of Science and Technology, 5, Avenue des Hauts-Fourneaux,
L-4362 Esch-sur-Alzette, Luxembourg
* Correspondence: jean-sebastien.sottet@list.lu
† These authors contributed equally to this work.

**Abstract:** Nation Wide Digital Twin is an emerging paradigm that pushes the context of a classical Digital Twin to a whole country. Under this perspective, models, which are central for digital twins, will play a key role for the design and implementation of such a specific digital twin. However, to achieve a nation wide digital twin vision, a whole set of problems related to models have to be solved. In this paper, we detailed the notion of nation wide digital twin with respect to well known digital twin from a model point of view and discuss the problems the community is facing in this context. As a result, from the identified challenges, we propose a research road-map paving the way for future scientific contributions.

**Keywords:** Digital Twin; models; data; Nation Wide Digital Twin

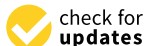



## 1. Introduction

The term Digital Twin (DT) first emerges from the manufacturing industry in the early 2000 as a digital replicate (i.e., a twin) of a production line or a product. If the notions behind Digital Twin are anchored in the early digital simulation environments, it has only been recently formalised by the original claimed authors of the DT definition [1]. DT aims at supporting any simulation [2,3] or prospective scenario such as predictive maintenance [4,5], in a digital world. Under the simulation, a DT should provide the same answer as its physical counter-part. According to [1], a DT is composed of three major elements: (i) a physical entity of the real world, (ii) its counterpart in the virtual world (the twin) and (iii) a bi-directional data exchange connecting those two worlds. In practice, it is hard to achieve this pure bi-directional exchange and most of the advanced works only consider the realisation of a digital shadow. A digital shadow [6] is a digital replica of the physical system[1] with few or even no possible feedback actions on the physical system.

As it has been defined in the digital era, data and data management infrastructures are at the core of DT concerns. Many works are focusing on the capture, mainly relying on IoT sensors, and aggregation of this Data. The main technological infrastructure are indeed dedicated to sensors and data management like FiWare [7] or Azur DT [8]. However, it appears in recent literature, that it should be complemented by human knowledge including designed artefacts, i.e., models [9,10]. Following this perspective, a DT is then perceived as a composition of heterogeneous sources of information comprising a huge variety of models including data models, design models, predictive and prescriptive models, semantic models.

Given the great impact DT have on the overall improvement of real world systems, this paradigm has spread beyond the traditional industry 4.0 use cases, reaching notably domains such as urban planning [11], energy consumption, green mobility, etc. Smart City and DT concepts are dealing with similar and complementary concerns [12]. When going beyond a smart city, considering for instance connected cities, regions or even a

country within all its aspects, we then speak about the concept of Nation Wide Digital Twin (NWDT)[2].

Initially proposed in Singapore and in the United Kingdom, NWDT are solutions to overcome the major systemic problems a country (or a large region, a big city) is facing and may encounter in the future: e.g., energy consumption reduction, natural risk prevention (like sea-level raising in Singapore [13]). The first incentive that justify the elaboration of a NWDT is the need to build an holistic view on how the country will behave in potential futures crisis and ideally predict those crisis before they happen. Even if they are inspired by classical DT defined initially in an industrial context, they require different approach, notably because the physical system borders cannot be a priori identified and is constantly evolving as well as the ultimate goal pursued by the NWDT. We thus consider NWDT as an open-world system. It is indeed very hard to grasp all the involved elements under the large umbrella of aspects a Nation could have. NWDT is then seen as a constantly evolving system that is built in an iterative manner.

NWDT covers many domains (e.g., economy, ecology, construction planing, etc.), and potentially aims at involving different stakeholders (e.g., country deciders, policy makers, engineers, citizens, etc.). Each concerns or domains addressed in the NWDT can be seen as smaller digital twins, also know as local digital twin [14]. They are potentially providing assumption at their border with other domains behavior (i.e., behavioral models). For instance, for the domain of transportation, a DT can consider estimation of the electricity domain: grid charge, green energy used, etc. As we see NWDT an open-world system, providing a precise but inherently restrictive definition of it will not serve our purpose. We rather see NWDT as a constantly evolving DT or federation of DTs that aims at answering the grand challenges of the country: e.g., how my country is able to face the crisis, what are the strong point, the weak points, how I can predict future issues related to climate change. It should also answer the deciders on the holistic impacts of a given policy, a new regulation, etc.

Smaller twins, composing a NWDT, can also be autonomous DT providing their own decision area (e.g., at the scale of a single machine, a quarter, a city, ...). Even at a smaller scale, there is a need of having a proper harmonization of the digitalisation (data and models) of the physical system. Under our viewpoint, NWDT is seen as a federation of different local DT at different scale and concerns. As a result operating a NWDT requires to handle a comprehensive and highly heterogeneous information; information being represented under the umbrella of the term model that comprises many definitions [15], classical design models, knowledge models, machine learning, behavioral data-centric,etc. As a result, to be as a close as possible to twin the physical system models, the NWDT should be supported by a proper model management infrastructure. This infrastructure should deal with the heterogeneity of nature of models, information granularity, semantic and their consistency to work as a whole.

Associating many transversal domains, stakeholders and their respective data and models requires dealing with a holistic governance adapting existing approaches such as federated systems [16] or distributed governance [17]. Indeed, a proper governance should preserve the sovereignty of information provided by stakeholders (i.e., models and data), quality and global relevance of provided information, as well as ensuring security and trust between the partners (mainly in the context of a commercial exploitation).

From our perspective, the quality of the governance will directly impact the way we align and federate the NWDT models together. A given model is potentially take precedence another (part of) model. We should also know who is able to access and operate which models. In this article we will not deeply study the governance and security aspects, but should keep in mind that governance is a key element for such as demonstrated by the Data Governance Act [18].

In this paper, we focus on the management and harmonization of the many data and models composing a NWDT. We believe that data and models harmonization, beyond the infrastructure supporting it, is the most important component of a NWDT due to its

intrinsic complex and variable nature. As it is by nature an open system, it also evolves through times, requiring the models to evolve with the different dynamic counter parts. Maintaining a NWDT is a constant process of refinement trying to fit the best with the physical system.

The remainder of the paper is organised as follow. We first introduce in Section 2 our understanding on data and model harmonization in the NWDT context. Then, in Section 3, we discuss the problem related of the management of models life-cycle (refinements, evolution) as the NWDT is progressively discovered and how to ensure consistency between models and NWDT. From those statements we highlight in Section 4 the resulting research challenges that the modelling communities have to face. Section 5 wraps up with concluding remarks.

## 2. Data and Model Harmonization in a Nation Wide Digital Twin

The notion of Nation Wide Digital Twin introduced before implies the involvement of several stakeholders having different contributions in the design and implementation of the twin. Modeling these inputs at NWDT levels will require a deep and clear understanding of the wide range of models that exists in current implementation of digital twins [19], their advantages and drawbacks. Actually, these models can be of different nature, type and can have different (and sometimes incompatible) formalism and semantics. This is the case for instance of the DT describing 3D printers in [20] which uses MTConnect as information model while the DT described in [21] uses proprietary ontology as information model to also describe 3D printer. This example underlines one of the many configurations involving models in DT that requires harmonization. As illustrated in Figure 1, models obtained from different artefacts (either data and models) in the real world requires some harmonization at different levels in order to be properly interpreted and jointly used. Potentially, different modelling languages and format are used, following either a standard or not. Indeed the different formalisation of data and model may cover similar concepts in different model or cover different sub-part of a given concept, they need to be harmonized to be exploited properly as a coherent whole. Finally the different (community) interpretations of those concepts may also lead to inconsistency that needs to be harmonized. In this section we will review and discuss specificities of a NWDT in terms of data and model harmonization.

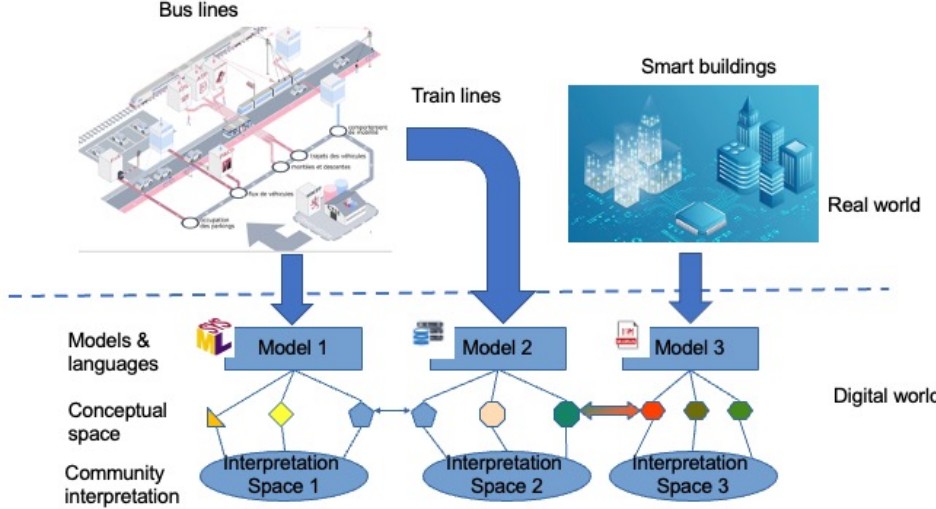

**Figure 1.** Data and models harmonization in a digital twin.

### 2.1. Standards vs. Proprietary Reference Models

The previous example suggests a first barrier towards the harmonization of models in the context of a NWDT: implementation of Standard versus Proprietary reference models. The majority of DT approaches advocated the use of standards because it provides a com-

mon understanding of the solution, an open access to the specification of the standards and sometimes bring together a community of users that can share their problems and solutions using standards (Figure 2 shows some examples of standards in the manufacturing domain). On the other hand, proprietary solutions, that can be seen as black-boxes (with limited access to the specification of the model), usually offer more functionalities than standards and dedicated support to implement the models in the DT.

In a NWDT context, this well-known problem can occur at different levels depending on the granularity of the elements that are considered and also on the point of view envisaged when developing the models. Actually, when comparing existing models for describing a resource (usually seen as the core entity for DT), whether they are standard or proprietary, it is difficult to establish a one-to-one mapping between their entities [22]. Consider for instance the central notion of "asset" in the Asset Administration Shell and the notion of "Interface" in the Digital Twin Definition Language (DTDL) developed by Microsoft. These two notions are supposed to describe a resource but their semantics are not equivalent. This is partially due to the point of view taken by these two models: the Asset Administration Shell focuses on resource description, resource discovery and resource access while DTDL only consider resource description.

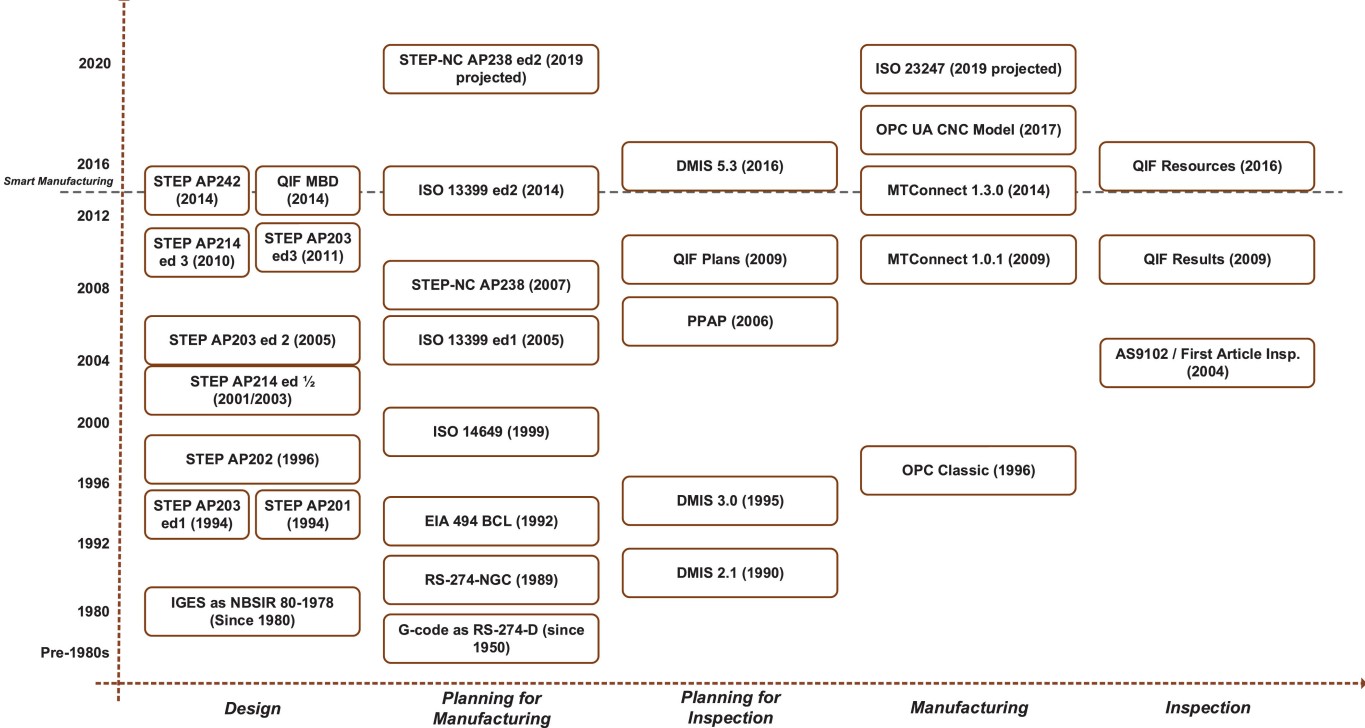

**Figure 2.** List of standards for describing physical objects in the manufacturing domain (taken from [19]).

Figure 2 gives also an illustration of the wide variety of existing standards in the manufacturing domain which is only a use case that can be covered by DT. Therefore, in a context of a NWDT, the significant number of standards that can be used also require a priori knowledge on their parameters, the way they interact, the type of handled data which demands harmonization to be fully exploitable.

### 2.2. Open World vs. Close World Assumption

The former section suggests another important characteristic of a NWDT that DT usually do not have: the open world assumption (deeply anchored in the Semantic Web domain). As opposed to close world (usually considered in classical relational databases systems), in an open world, what is not known cannot be considered as false. In classical DT, the twining usually occurs on an overall system, e.g., a rocket, a car, production line,

etc. with a clear understanding of the context and the interactions between this context and the system are known and predictable: the system is then considered as complete as well as its DT meaning that the close world assumption applies.

In a NWDT paradigm, the system to deal with is usually too complex and shared between several entities therefore cannot be fully known. Consider for instance the management of public transport at country level. Coupling the information captured by the geo-localisation of a bus and its electric consumption (known by applying the close world assumption) you can easily verify if the bus is following its planned schedule. However, if the traffic jam forces the bus to deviate from its route (taken into account under the open world assumption), this might impact its energy consumption but also all the schedule of other buses or maintenance tasks that have to be performed on that bus. On this example, the interaction with the environment (i.e., what is outside the current scope of DT like the traffic jam) is not fully predictable. Moreover, DT requires a function that make the link between the physical and digital worlds mainly to inject data acquired in the physical system into the models composing the digital twin. However, this data injection is not always done in real-time therefore the adoption of the close world or open world paradigm drastically impact the system since considering the missing data as false would have different impact than considering that information as unknown.

Considering the models involved in the design of the system following a model-driven engineering perspective under the open world assumption means that the models of the system are potentially incomplete giving only a partial view on the system. This is also true at data level since, at national level, it is rare that an organisation own all the necessary data needed to mimic a real system like public transportation. For instance, the infrastructure providers have topological and geographical models of lines (e.g., bus, railways) and the dynamic view on traffic flow (e.g., trains position); public transport authorities have business processes and architecture models; whereas transport means operators may only have time schedule and paper-based information (e.g., drivers schedules). As a consequence, only a consistent subset can be involved in operations, whereas other parts have to be inferred either from known data or from other existing model elements. It will require a modelling environment that will be able to handle both fixed and inferred elements whilst ensuring the possible operations on models [23]. Moreover, the modelling environment must be open and flexible enough to be connected with potentially new aspects that can come through the progressive discovery of the physical system. Under the open world assumption, this flexibility is allowed (i.e., taking into account a completely new kind of event) but not under the close world assumption. This underlines the need to consider uncertainty in the models [24]. Uncertainty will support the representation of the potential occurrence of a fact or a partially defined element of the system than must be represented on a twin. In order to be able to implement automatic reasoning over the twin (e.g., to perform simulation) some assertions must be taken: either closing a view on the DT at a certain point in time, making assumption on incomplete elements and/or being able to evaluate the uncertainty consequence.

### 2.3. Static vs. Dynamic Models

Static models are usually a category of models that describe factual knowledge based on the concepts and relationships of a given domain. In this category we can cite: fact-based models, evidence-based models from the medical field, conceptual models such as ontologies [25], object role models [26] or data schema. On the contrary, dynamic models are the kind of models that represent dynamic knowledge which includes processes, description of behaviours, executions or more generally speaking the transformations of a given system. The "dynamic" adjective puts therefore the stress on the nature of the knowledge that is represented using these models and not on the behaviour of these models themselves (see Section 3). This is why dynamic models can be used either to describe a behavioral aspect of the system in a static manner or actually "operationalise" the models to make them executable. Dynamic model can be seen as a set of functions that describe the

change from one state to another. Dynamic models can have many forms and comprises for instance state transition diagrams, Ordinal or Partial Differential Equations (ODE), encoded functions, etc. In a NWDT context, both family of models (i.e., static and dynamic) coexist and interact with each other. Static models such as ontologies can, for instance, be used to represent the core knowledge of a system through a well-defined conceptualisation associated with a formal semantics or it can describe the input and output of a process. Ontologies also offer the possibility to describe data that is used in a DT via metadata [27] however, their inability to represent dynamic knowledge limits their usage. This is why, ontologies are mainly used to bridge the gap between the various categories of models reviewed in this section as highlighted in [28]. They act as a reference model that allows to link, from the semantic point of view, the various concepts expressed in the set of models implemented in a DT whether they are of static or dynamic nature. In this case, maintaining ontologies on top of models allows for providing a common vocabulary for the different models and stakeholders whatever is the modelling language and formalism they will use. The property of a model to describe the change that impact other models can have many shapes. It can be a function that can e.g., simulate a static model change through time (i.e., change the model state), or a model of a transformation that convert a model in another one, or change a model in its evolved (future) form.

## 3. Model Life-Cycle Management and Consistency Checking

Since a model is only an approximation of the reality, it is by definition, subject to frequent change in Digital Twin context. Indeed, different operations on model elements, even relying on AI, can be used to make the models being closer to the reality or to express discrepancy with the reality [29]. As shown in the previous section, the NWDT demands the harmonization of the many different underlying models. It also requires some flexibility, especially if considering NWDT and the physical system as an open world system. The consequence is that such a system will constantly evolve over time: in such an dynamic environment, dealing with consistency is challenging. We have to manage the consistency during this evolution on the one hand, and, in the other hand, to deal with the flexibility of models which may results in necessary inconsistencies (e.g., due to the different viewpoints).

Based on our understanding, we propose the three following assertions that show the need for managing the evolution of NWDT models all along its life-cycle:

1. By virtue of real world evolution, we consider that the physical system is constantly evolving—even if some parts of it remain very stable over time. As the perception, political orientation, citizen concerns and, event impacting the world change over time.
2. It is hardly possible to have a clear and complete view of a country and therefore, the construction of a complete NWDT at once is hardly reachable. As a result, a NWDT will progressively be refined through time.
3. In practice, we will not consider a whole encompassing NWDT but rather the connection of existing local or domain DTs to form a federation of DTs. This requires the consideration of a distributed model approach, with restricted access to the model elements (as they will remains the property of third party entities).

The previous assertions suggest that consistency in models, their environment and the underlying tasks is cornerstone for NWDT. This is why, methods for ensuring this consistency are necessary to ensure harmony between models. This is the case, for instance, when connecting a new external DT to the existing NWDT, it will be necessary to ensure that information is consistent, notably if one DT is overlapping with the others (in terms of input or output) to make proper assumptions on it. We should also keep in mind, that at certain moment in time it is necessary not to be too rigid regarding this harmonization to keep the different possible interpretations (viewpoints) of the physical system. We should however trace those inconsistencies and possible mismatch of models that allows for, taking a specific viewpoint, some reasoning over the NWDT models. We illustrate those different

situations in the following sub-sections introducing, for each of the aforementioned points, the related challenges.

### 3.1. Supporting the Evolution of the Physical System

As previously asserted, in a NWDT we must consider that its physical part will constantly evolve over time, e.g., new building in a city, new transportation line, etc. As such, the NWDT models have to evolve through time to cope with the reality: like we need to maintain requirements for an evolving software [30]. We also must consider that the knowledge and perception about the system is also changing through time. It can be a drastic change, in the perception of system or precising some elements of the systems or on the contrary, have doubt about the model relevance (and add uncertainty into it). Indeed, we must keep in mind that models are representation of system under a stakeholder point of view. It leads to a subjective view on the system which can be discovered as erroneous during the NWDT life-cycle. Similarly to classical Digital Twins, the NWDT will also rely a lot on means such as sensor and IoT devices in order to inject data and instantiate models. Indeed, IoT devices (notably their communication, quality and type of information) are also dynamic entities that will potentially force the perception on the physical system to evolve and consequently will impact the NWDT.

If the evolution of a system can manifest itself in several ways, we can summarise the main different causes of evolution of models in the following enumeration:

- The physical part of the NWDT changed. This result in changing a part of the models of the digital part that correspond to the identified changes in the real world.
- The model is not aligned to reality due to bad modelling work. This is usually detected at consistency checking or at run time when analysing the behavior of the NWDT.
- The perception of the reality coming from the sensors or given by the humans has changed. This can result in a drastic change of the NWDT models because they are no longer aligned with the perceived reality. This can also reveal a drift in the semantics of the element of the models (see Section 4.2), which lead to precising some part of the model or on the contrary to add uncertainty.

Due to these frequent changes, it is necessary to have a model maintenance process to ensure that the DT models, and depending artefacts are not deeply impacted by the evolution of the physical system. Such a maintenance approach has to be designed independently to the nature of the implemented models including numerical models issued from Machine Learning approaches and AI in general. This process can be inspired, for instance, from the Knowledge Engineering community where Bryce et al. [31] identify deviation between the knowledge model as formed by the stakeholders, the reality (the physical system) and the formal models describing it. Theses three elements (knowledge, physical system and representations) have to be aligned if we want to ensure consistency either to the reality and also between the different perception of stakeholders.

In such unstable environment, models should be flexible enough to be adapted to change and representation deviation, being potentially completely refactored. The same applies to any piece of knowledge represented via ontologies or knowledge graphs associated with the models. As a summary, being able to manage models in such a changing environment requires to have flexible representations of the system, i.e., flexible modelling artifacts including knowledge representation. It consequently requires to have a maintenance support in order keep models up-to-date and consistent together and with the reality. Indeed, this process can be related to a continuous building of the NWDT as a complex system.

### 3.2. Continuous Building of Nwdt

Building a NWDT, or even a DT at a smaller size, like for a region or a city, is not fully graspable in one modelling session. On the contrary a NWDT must be built incrementally involving stakeholder's knowledge, consensus found amongst them but also based on the data that are available and that allow to mimic the physical system. Under this perspective

a NWDT is under constant construction, the NWDT will only give a partial representation of the reality, potentially exposing as uncertain some elements not already fixed, not well known or not even represented. As such, embedding uncertainty, as assumption on potential facts of the reality, in the models is required.

Uncertainty has been widely studied in modelling. In their paper, Troya et al. [32] show the diverse forms of uncertainty that exist at model level and the way to represent them. Traditionally, uncertainty is divided into two categories: aleatory and epistemic. The former is said irreducible, in the sense that the information to solve it or to deduce it is not available. Indeed it should be kept as a model element in the NWDT as the representation of an aleatory phenomenon on the physical system or imprecision of the sensors' measure on the system (e.g., represented as range of possible values). The later category denotes the epistemic uncertainty that refers to the lack of knowledge of the system which is then reflected into the models. In our view of continuous building of the NWDT, we envision that a process will progressively reduce (or eliminate) this kind of uncertainty over time. Moreover, epistemic uncertainty can also be used to progressively find a consensus between the various actors when building a NWDT [33].

When dealing with such a organic growth of the NWDT system we need to keep a track on the state of each of those model elements composing it: what is defined, what is unsure, what has to be defined, what are the source of information (i.e., data sources, sensors and their quality/confidence), etc. This can be, for instance, represented in a knowledge graph that will be used as an integrative backbone of models during all the life of the NWDT.

### 3.3. Connecting External Models from Other DT

Considering that every domain pertaining to a Nation has not already been studied will be foolish. Indeed, in specialised domains such as the manufacturing industry or logistics some DT already exists (or without being a full DT, it is already a precise digital model). A NWDT should therefore be able to ingest the information coming from other DT or any complex source of information (e.g., the digital model of public transportation in the country). Technically those interoperability problems can be solve by relying on standard interfaces solutions or exchange models. In the manufacturing industry, co-simulation & the Functional Mockup Interface (FMI) standards are tackling the problem of exchanging dynamic simulation models by providing the appropriate mechanisms. FMI defines a container and an interface to exchange dynamic simulation models using a combination of XML files, binaries and C code, distributed as a ZIP file. Nevertheless, all the domains of the NWDT are not as mature as co-simulation in industry and may have no standard description for interoperability purpose.

In addition, for managing this multi-source DT, even within well-defined model exchange conventions, a proper governance will be an important aspect to deal with. Also, a peculiar attention should be put on privacy and sovereignty of data, models and results of computation processes. Existing research already provide some concrete best practices [34] or data infrastructure and regulation such as Gaia-X [35] that could inspire a NWDT approach, peculiarly regarding the governance over models. Moreover, approaches like OPAL (OPen ALgorithms) [36] could also be of use: as they provide solutions to enhance trust in algorithms that fetch data from other sources but allow the data owner not to communicate all its data.

Considering that the full DT models coming from a third party remains the property of this third party, building a single unified aggregation of all local DT in our NWDT is hardly feasible in practice. Approaches dealing with a centralised unified model, like SUM [37] will not be relevant in that context. We should rather consider the communication of models (and data) as potentially incomplete (i.e., what is communicable form a third party, what is relevant from a legal viewpoint, possibly transformed model). As a result, approaches like Distributing models as fragments [38] is an interesting solution if adapted to a distributed DT architecture [39]. Indeed, it will help to communicate only the useful

piece of models (i.e., model fragments) that are required (e.g., in the context of privacy, removing all personal information) and relate together models from different sources. This distribution of models could not only help with a global perspective, but should also help local decisions [14].

## 4. Research Challenges

The notion of NWDT has recently emerged this is why several research challenges have raised and deserve to be investigated. In this section, based on the content of previous sections, we propose a list of research problems and clues to approach these issues.

### 4.1. Model Reconciliation/Semantic Mismatch

As NWDT concept relies on models, their reconciliation is de facto the main issue. This can take the following forms:

- Models have to be properly **documented** and **the semantics** of the different elements composing them **must be explicit** to have a clear, deep and unambiguous understanding of the models. In classical DT approaches, software developers have the global overview on the various models involved which reduces drastically the bad interpretation of the content of the models. However, in a NWDT context, specific problems raise. On the one hand, as evoked in Section 2.1, elements from different models can have the same labels but have different semantics because of the different views one can have when building the model or simply because the models have totally different context or even because of semantic drift issues (see Section 4.2). On the other hand, elements of different models can have different labels but the same semantics and therefore must be properly aligned to be used. Current works approach this problem via knowledge graphs. This is the case of [40] where the ability of knowledge graphs for knowledge representation and the dynamic of agents to update the graphs in near real-time are considered for a NWDT focusing on energy saving. In [41], the authors addresses the design of a complex DT design where classical DT are augmented with much more elements, the variable scale of working environments and changeable process among others. In their approach, they use ontologies to establish the complete information library of the entities on different digital twins and knowledge graphs to bridges the structure relationship between the different scales of digital twins.

- **Models evolution needs to be properly managed** to ensure a continuous exploitation of the DT. In a NWDT context this is even more difficult since different models, evolving at different speed, are implemented. As evoked in Section 3, the evolution of models have first to be captured and characterised this means that we need to know what has changed in the model and what kind of changes occurred (e.g., atomic changes like addition or deletion or a combination of atomic changes like split or merge). The complexity of computing this diff is highly depending on the nature of the model (e.g., graph based [42]) and on the formalism used to express these models. This phase is crucial for another important aspect that consists in propagating the changes to depending artefacts (i.e., other models composing the DT) [43,44] which requires a proper governance of the NWDT (see Section 4.4).

### 4.2. Semantic Drift

In the semantic web, semantic drift refers to "how concepts' intentions gradually change as the domain evolves" [45]. Following this definition, the major problem is the detection of this drift because this conceptual evolution is slow and may be the result of the change of the model itself (intrinsic drift) or the result of the change of the real world and in this case may require the analysis of data an information that are external to the considered model (extrinsic drift). This problem is intensively studied. In their work, [45] introduces a method to detect semantic drift in ontologies by analysing the signature of concept. The proposed signature consists in a mapping between the concept label and linguistic expression that are used to denotes the considered concept. They construct vectors of

signature for different version of the ontology and then measure the similarity between these vectors to detect the drift. Similar approach is presented in [46] but the notion of morphing is introduced and used to detect semantic drift using the label, intention and extension of concepts therefore covering the intrinsic and extrinsic aspects. The SemaDrift framework has been recently developed to implement existing approach for detecting and visualizing concept drift in ontologies [47]. As our focus is NWDT, it is important to survey methods addressing **the near "real time" aspect**. This is the problem addressed in [48]. Their detector relies on computing multiple model explanations over time and observing the magnitudes of their changes. Social media is also a source of data that is used to detect semantic drift [49] via classification methods. In the context of a Model-Driven Digital Twin where several models interact, an important problem concerns **the propagation of the identified drift** to dependent artefact (i.e., other model entities). For instance, when several heterogeneous models are aligned, when a remapping is required after concept drift and how to remap the involved concepts? To answer this problem, to the best of our knowledge, no approach has been developed. The only tentative is described in [50] where the authors evoked the use of knowledge graphs to provide the semantics of model entities in a Cognitive DT context and the ability of logic-based reasoning techniques to identify semantic drift and manage its impact on depending artefacts. However, Kalaboukas et al. only discuss at a very abstract level this issue but do not addressed it concretely. In our recent work, we have addressed the problem of the propagation of model evolution to mappings [43] and semantic annotations [44]. However, this has been done in a pure Semantic Web context and not in an environment in which heterogeneous models coexist. In a digital twin context, concept drift detection and propagation have been addressed at the data level not at semantic level [51]. In DT applications, concept drift refers to the degradation of time series treated by DT models for various tasks such as prediction [52].

*4.3. Model Flexibility*

A NWDT can also be seen not only as a large all-encompassing DT for the nation but rather as a federation of specialised DT each designed for a specific sector or domain of activity in a given country. For instance, financial service of a country will provide a DT whilst other services like transportation, health, etc. will provide their own DT which will result in the cohabitation of a set of DT and can therefore be seen as a system of systems [53]. Under this perspective, **models have to be flexible** to facilitate their integration, alignment or combined together to have a relevant impact. Consider for instance a sensor sending data to the DT: models in use in the DT must be flexible enough to (i) **be easily modified to support a too important drift** of the values send by the sensor with respect to the corresponding element of the model and (ii) **to tolerate a minor drift** of the mentioned values. Other example of flexibility is considered for the ontology alignment task [54]. Even if concepts of an ontology are fully aligned, it is only few attributes of these concepts that define the mapping. This example indicates a loose consideration of the semantics of the elements with respect to the impact the mappings could have on the decisions that can be made using the ontologies. Flexibility has also been addressed by the MDE community where, for instance, Ref. [55] propose a framework that support the relaxing of standard metamodel definition in order to express uncertainty and unforeseen modelling constructs. As evoked in [32,55], this flexibility also takes **uncertainty at model level** into consideration (see Section 2.2 on the open world assumption). In this context, recent approaches have used Fuzzy logic [56] or Hankel matrices [57] generated from input and output data to deals with this uncertainty. Usually, this uncertainty is measured via probabilities which need to be frequently revised because of the dynamic nature of the physical part of the DT and our comprehension of it which must be pushed to its digital counterpart.

*4.4. Security and Governance Aspects*

In the context of any information system, **the management of information, data and models is crucial** and requires a proper governance. The NWDT does not escape this rule. From a technical point of view, the governance is challenging since many different technologies will be involved such as the many IoT sensors, AI processes, local DT architectures, communication channels (components and protocols), etc. In this context, data and model harmonization will play a major role notably through the aforementioned concepts such as identifying semantic drifts and mismatches but also being able to provide flexible models.

Secondly, from a content-wise perspective, the governance of models (including data) is even more challenging. The model and data harmonization to be realised requires to **be really precise in terms of ownership**, availability and permissions to access/use data and models. This issue, beyond data ownership, is indeed addressed in modern data regulations (e.g., GDPR[3] or in framework like Gaia-X [35] in the E.U.). As a result, the two following main governance concerns arise:

- Ensuring the privacy (Transparency, Unlinkability and Controllability) of the processing of data.
- Ensuring the sovereignty of the data controller to decide on the terms of use of data by the involved third parties

**Privacy** is a challenging domain for DT [58] because of the many sources of (personal) data and the aggregation of multiple information necessary for the NWDT. For instance, citizen personal information can be spread in many domains and a cross-checking of those information may lead to identity theft. In addition, due to the variety of components, there are many sources of potential data leaks that harm privacy [59]. Note that, conversely, a NWDT can be used as a governance tool for a given domains (e.g., for the wind energy production [60]). Modelling of the privacy, security and governance aspects are part of the NWDT, directly impacting the country (e.g., law enforcement) and be a support for political deciders.

**Security** is an additional crucial aspect to consider. Indeed, a NWDT, is of interest for hackers, as a system of systems, that encompasses potentially all the country's information. A NWDT is indeed very hard to secure, if we consider it under a federation of local DT this can become even worse due to the variety of software, hardware and actors. Each local vulnerability can be an entry point to the overall NWDT. A compilation of specific threads and their impact on the DT [61] shows that standard security risk analysis can be applied to the different component of the NWDT with a stronger impact on the cascading risks. Due to the complexity of NWDT, it claims for an automated, dynamic and synchronized (amongst collaborating DTs) [62] security operations. Modelling all the governance and security concerns that affect the NWDT, and more peculiarly at the level of the models and data pertaining the NWDT themselves will help to hold the complexity required by such management.

**5. Conclusions**

Nation Wide Digital Twin is a new paradigm that has gain in importance over the past years through the UK and Singapore initiatives. Even if it relies on the basic notion of Digital Twin, it has some major differences which generate new scientific issues. In this paper we have discussed specificities of NWDT from a modelling perspectives and have identified the main research challenges that have to be addressed in the coming years. In our studies, it turns out that the dynamic aspect of digital twin, extended at country level, is causing new families of challenges which includes (i) the reconciliation of heterogeneous models from the semantic perspectives, (ii) the proper management of the evolution of these models and the propagation of changes that occurred in the real world to the corresponding models and then from these models to depending ones, (iii) the flexibility of the models to ensure a proper and smooth continuity in the performance of the NWDT and last but not least, (iv) the governance of the NWDT, notably when several local DT are involved and interacting with each others. In our recently funded project Model-Driven Digital

Twin - Semantic Drift (MDDT-SD), we will focus on these aspects in order to propose new methods to address these issues.

**Author Contributions:** J.-S.S. and C.P. have contributed equally to document. All authors have read and agreed to the published version of the manuscript.

**Funding:** This research received no external funding.

**Data Availability Statement:** Not applicable.

**Conflicts of Interest:** The authors declare no conflict of interest.

## Notes

[1]　A physical system is referred to as the actual system under study and may also cover its digital aspects.

[2]　We can call it a XWDT where is X can be replaced by the scope of the DT, e.g., RWDT for region-wide DT; our the ultimate goal is to consider holistic problems at a nation level.

[3]　https://eur-lex.europa.eu/eli/reg/2016/679/oj (accessed on 12 December 2022).

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
