# Peer review of "Data and Model Harmonization Research Challenges in a Nation Wide Digital Twin"

_systems, doi:10.3390/systems11020099_

Round 1

Reviewer 1 Report

The paper addresses challenges in the harmonization and management of data and models in the context of Nation Wide Digital Twins (NWTD).  It is a survey-like work with emphasis on researching challenges.  The challenges are categorized into four kinds: (1) model reconciliation/semantic mismatch, (2) semantic drift, evolution management, (3) model flexibility, and (4) security and governance aspects.  

Regarding originality and contribution to scholarship, I classify the paper as average given its survey nature. The challenges discussed are main concerns in the various areas of information system interoperability, model driven engineering and system engineering in general, as acknowledged by the authors from the citations in the manuscript.  

Nonetheless, in the context of the Special Issue "Digital Twin with Model Driven Systems Engineering", I think that discussion in the paper can be profitable the researchers in the area.  

Regarding the form, the text is well written. Some minor issues are:

Page 1, footnote 1: "... refers to [what??] as the ...".

Line 52: "They potentially providing ... [??]". 

Line 77: "... governance ans [??] security ...". 

Line 215: "Based [on??] our understanding ...". 

Line 229-30: "... methods for ... is [are??] necessary ..." 

Line 252: "... new connections/new elements [??] ..." 

Line 253: "... forces [force??] ..."

Line 294: "Uncertainty [h]as been ..." 

Line 339: "... emerged [??] this is  ..."

Author Response

Dear Reviewer 1,

Thank you for your review. As you said our paper is more setting the ambition of the Nation-Wide digital twin from a model harmonization perspective, and as you said, it is more oriented for discussion around our research on this subject.

We will do the typos and grammatical correction in the modified version of the paper to be submitted.

Best regards,

Reviewer 2 Report

Digital twin is an emerging and important research field. This paper provides a good review of the nation-wide digital twins. The structure of this paper is reasonable. I only have a few minor issues that need to be revised before publication. First, it is recommended to better explain Data and Model Harmonization in the form of a diagram. Then, it is recommended to review the following two digital twin papers(10.1016/j.ymssp.2022.109896 and 10.1109/TMECH.2022.3218771).

Author Response

Dear Reviewer 2,

Thank you for your review. We have modified the document according to your comment. We have produced a diagram that explains the different data and model harmonization from our viewpoint: combining concepts, models and languages as well as the ontological space. We also study referenced papers that reflect some of the discussion we have on dynamic modelling and lifecycle.

Best regards,

Reviewer 3 Report

The paper introduces some interesting new aspects regarding the models required for the new concept of Nation Wide Digital Twins (NDWT).  The arguments are sound and the exposition is good, although the style is more that of an essay than a technical / scientific paper.

The paper could be improved by addressing the following points:

1. A solid definition of NWDT.  There is no definition in the paper, just a description and reference to [11]. At least the scope, or a minimum scope should be required. With the current definition, the NWDT could be anything or nothing.

2. The definition should include the expected functionality. Otherwise arguments such as lines 176-179 seem quite arbitrary.

3. The definition of "dynamic" should be compared to that used in Dynamic Systems (i.e. systems of ODEs or PDEs). In general it is left unclear what level of integration the various models should enjoy, or what interfaces might exist between them.

4. Line 286: Why are there frequent changes to the model? Perhaps it would help to introduce a lifecycle model for the NWDT.

5. Lines 326-337: Such distributed or federated models are also addressed by Co-Simulation & the FMU/FMI standards. This should be referenced.

6. The overall goal WHY someone would want to build a NWDT should be addressed.

7. A table of the key requirements / characteristics of the NWDT should be included.

Author Response

Dear Reviewer 3,

We have acknowledged your review. We perfectly agree that this paper aims at putting the emphasis on Nation-Wide digital twin under the viewpoint of model and data harmonization, which is, according to us, rarely studied; in this context we believe that even if the style is more an essay it as an interest for discussion since it has different potential research aspects regarding the traditional close digital twins systems.

  1. We have proposed to describe the properties of a NWDT in the introduction, we have thus added some sentences to precise more it has a definition. We put in red the paragraph that describes the properties.
  2. We have added the possible usage and functionalities that a NWDT should provide in the introduction in relation with the definition provided.
  3. We did include the considerations about ODE and PDE (in red, line 206 and further) which are part of the dynamic models that we should consider, we also depict other forms of dynamic models that should be harmonised.
  4. By definition, a model approximates the reality therefore changes at model level is needed to bring the model closer to the reality. These changes can be induced by a modification of the real world, a bad conceptualisation of the model or a new formalization (among others). As an example, an AI model need to be frequently trained with new data to reflect the reality. We modify the text to make this clearer (see section 3)
  5.  We have considered FMI (in red from Lines 340 to 348), as a way to perform alignment of dynamic part of the models; as we stated this is a standardised way to do it, but in the context of a NWDT, the level of maturity should be not sufficient to have already standardised interfaces.
  6. We try to make it clear the goal of the NWDT and what are the potential interest for governmental organisations to consider the country as a system to be twined.  We have highlighted that in the introduction. We try to make a clearer explanation from line 39 to 49, notably illustrated by the examples of the UK and Singapore.
  7. Having a table will be completely disconnected from the article structure, we rather propose to highlight in bold the key characteristics of the NWDT related to potential research (see section 4).